

# Walking on different inclines affects gait symmetry differently in the anterior-posterior and vertical directions: implication for future sensorimotor training

Haoyu Xie[1,2] and Jung H. Chien[2,3]

[1] Department of Rehabilitation Medicine, First Affiliated Hospital of Sun Yat-Sen University, Guangzhou, Guangdong, China
[2] Department of Health & Rehabilitation Science, University of Nebraska Medical Center, Omaha, NE, United States of America
[3] Independent Researcher, Omaha, NE, United States of America

## ABSTRACT

A symmetric gait pattern in humans reflects near-identical movement in bilateral limbs during walking. However, little is known about how gait symmetry changes on different inclines. This study aimed to address this knowledge gap using the central pattern generator and internal model hypotheses. Eighteen healthy young adults underwent five 2-minute walking trials (inclines of +15%, +8%, 0%, −8%, and −15%). Dependent variables included step time, step length, step width, maximum heel clearance, time to peaks of maximum heel clearance, their corresponding coefficients of variation (CV), and respective symmetry indices (SI). Significant differences were observed in SI of step length ($p = .022$), step length variability ($p < .001$), step width variability ($p = .001$), maximum heel clearance ($p < .001$), and maximum heel clearance variability ($p = .049$). Compared to level walking, walking at −8% and −15% inclines increased SI of step length ($p = .011$, $p = .039$ respectively) but decreased SI of maximum heel clearance ($p = .025$, $p = .019$ respectively). These observations suggested that incline walking affected gait symmetry differently, possibly due to varied internal models used in locomotion. Downhill walking improved vertical gait symmetry but reduced anterior-posterior symmetry compared to level walking. Downhill walking may be a preferable rehabilitation protocol for enhancing gait symmetry, as it activates internal model controls. Even slight downhill inclines could increase active control loading, beneficial for the elderly and those with impaired gait.

# INTRODUCTION

Walking is a phenomenon that can exhibit both simplicity and complexity, regardless of whether it is observed in healthy populations or those with pathological conditions. For instance, a study reveals that eight out of fifteen stroke survivors (over 6 months over stroke)

Corresponding author
Jung H. Chien,
drjc.science@gmail.com

could walk over 80 cm/s on the level ground compared to 100 cm/s for healthy controls (*Stern & Gottschall, 2012*). However, researchers also observe the changes in the gait pattern when they are going uphill or downhill compared to level walking, indicating that walking on the level ground seems like a relatively simple task. These changes have been identified in young adults by reducing the step length when walking downhill (*Xie, Liang & Chien, 2023*), in older adults by taking slower and shorter steps during transitions between level and downhill or uphill surfaces (*Sheehan & Gottschall, 2015*), in toddlers by increasing gait variabilities (*Stern & Gottschall, 2012*), and in stroke patients by reducing walking speed and step length when walking downhill (*Phan et al., 2013*). Two main theories, the central pattern generator (CPG) and the internal model, explain this complexity. CPGs coordinate rhythmic movements without sensory feedback, as shown in studies on animals (*Brown, 1911*; *Brown, 1912*; *Shik & Orlovsky, 1976*) and people with spinal injuries (*Dobkin et al., 1995*). Surprisingly, some patients with Parkinson's disease struggle to walk but ride bikes smoothly (https://www.youtube.com/watch?v=aaY3gz5tJSk&t=31s), supporting the CPG theory (*Tiihonen et al., 2021*).

Although CPG can produce repetitive movements, maintaining the stability of these movements across diverse tasks requires an internal model (*Ryu & Kuo, 2021*). For instance, passive dynamic walking, akin to a CPG-like gait, showcases a human-like gait but lacks stability, emphasizing the need for an integrated approach combining the internal model and CPG (*McGeer, 1990*; *Ryu & Kuo, 2021*). This internal model encompasses neural mechanisms that synthesize information from sensory modalities and predicts movement outcomes, as exemplified in tasks like gripping a ball (*Kawato, 1999*; *Pierella et al., 2019*; *Merfeld, Zupan & Peterka, 1999*). The vision estimates the ball's weight (forward model), enabling the person to generate initial grip forces based on previous experiences. If the ball's weight differs from the estimated weight, adjustments will be made in the subsequent attempts when catching the ball until optimal grip forces are achieved (inverse model). The internal model hypothesis finds further support in studies investigating upper limb movements under external forces, revealing adaptive trajectory adjustments over repeated attempts (*Shadmehr & Mussa-Ivaldi, 1994*). Similarly, in split-belt treadmill walking paradigms, initial gait variabilities of gait symmetry decrease over time through continuous adjustments by the internal model, highlighting its role in adapting to novel locomotor tasks (*Mawase et al., 2013*; *Ogawa et al., 2015*; *Reisman et al., 2007*). While CPGs provide rhythmic gait, the internal model collaborates to mitigate tripping risks, as seen in treadmill walking studies (*Dzewaltowski et al., 2021*; *Mawase et al., 2013*). *Darici & Kuo (2023)* demonstrate how encountering unexpected uneven surfaces prompts deliberate adjustments in walking speed to maintain balance, underscoring the role of internal models in regulating locomotor states. This deliberate variability in step-by-step adjustments reflects an organized response rather than random noise, highlighting the intricate interplay between the internal model and locomotion control (*Darici & Kuo, 2023*). In summary, daily locomotor tasks, whether on inclines or uneven surfaces, require both the internal model and CPG to ensure stability and adaptability in movement (*Darici & Kuo, 2023*; *Ryu & Kuo, 2021*).

Research shows that during walking on inclines, CNS gathers information from the somatosensory, visual, and vestibular systems to maintain balance (*Xie, Liang & Chien, 2023*; *Sun et al., 2023*). Uphill walking increases stride length, duration, and muscle engagement compared to walking on flat surfaces (*Franz & Kram, 2012*; *Kimel-Naor, Gottlieb & Plotnik, 2017*; *Sun et al., 2023*; *Wall-Scheffler et al., 2010*; *Xie, Liang & Chien, 2023*), while downhill walking reduces these parameters (*Franz & Kram, 2012*; *Xie, Liang & Chien, 2023*). Additionally, uphill and downhill walking increases spatial-temporal gait variability (*Xie, Liang & Chien, 2023*). These changes align with the internal model hypothesis, where CNS adjusts iteratively during treadmill walking on inclines (*Ito, 2008*). CPGs generate cyclic movements, while the internal model controls these movements concurrently. This prompts questions about gait symmetry during uphill and downhill treadmill walking and its relation to the internal model hypothesis.

Over the past century, multiple studies have documented gait asymmetry in various parameters such as stride length, foot placement angle, range of motion, muscle strength, ground reaction forces, and muscle activations in healthy individuals (*Chodera, 1974*; *Damholt & Termansen, 1978*; *Herzog et al., 1989*; *Ounpuu & Winter, 1989*; *Stefanyshyn & Engsberg, 1994*). Recent research suggests that gait asymmetry may start in middle-aged adults (*Bonilla Yanez et al., 2023*; *Casabona et al., 2022*; *Laroche, Cook & Mackala, 2012*). Sensory deterioration may contribute to sensory reweighting and adjustments in the internal model (*Barra et al., 2010*). Limb dominance also affects gait asymmetry, with the dominant leg primarily stabilizing gait (*DeVita, Hong & Hamill, 1991*; *Hirasawa, 1979*). While walking uphill, swing time between legs remains similar to level walking, but downhill walking doubles the asymmetry rate, though not significantly (*Kimel-Naor, Gottlieb & Plotnik, 2017*). Also, stride time variability shows no significant difference between walking conditions (*Kimel-Naor, Gottlieb & Plotnik, 2017*).

Gait symmetry is commonly assessed using the symmetry index (SI), where values close to zero indicate symmetry, while values approaching ±100% indicate increasing asymmetry (*Kaczmarczyk et al., 2017*; *Patterson et al., 2010*; *Robinson, Herzog & Nigg, 1987*; *Sadeghi et al., 2000*). However, this calculation might have a potential limitation that if half of participants demonstrate the SI of −20% and the other half have the SI of +20%, the mean SI value would be zero, indicating a perfect gait symmetry. Thus, a study proposes modifying the SI by adding absolute function on the difference between two legs. This modified SI has been used to investigate the gait symmetry in healthy adults (*Stacoff et al., 2005*; *Van Drongelen et al., 2021*), patients with unilateral hip osteoarthritis (*Schmidt et al., 2017*), patients with bilateral hip osteoarthritis (*Van Drongelen et al., 2021*), and patients with stroke (*Wang & Bhatt, 2022*). These studies indeed reveal the preferred leg (not necessary be a dominant or unaffected leg) reliance (greater SI values) in aforementioned pathological groups during different locomotor tasks.

Hence, examining the symmetry of spatial-temporal gait characteristics while walking on different uphill or downhill inclines is imperative because even a slight change in incline during treadmill walking may lead to step-by-step adjustments in the internal model, as suggested by previous research (*Khandoker et al., 2010*). Addressing this knowledge gap is essential for enhancing our understanding of the internal model governing human

movement control during walking on different inclines, achieved through adjustments in gait symmetry. To test this internal model hypothesis, participants recruited for the current study had no prior experience with walking uphill or downhill on a treadmill. This study proposed the following hypotheses: (1) walking on an uphill or downhill incline was expected to result in a significant decrease in gait symmetry (greater SI values of spatial–temporal gait characteristics), indicating the preferred leg reliance due to changes in the internal model; (2) walking on different inclines (uphill or downhill) was expected to show significantly less symmetry of gait variability compared to level walking, indicating that greater gait variabilities might be occurred in one leg than another leg.

## METHODS

### Participants

A total of eighteen healthy young adults participated in this study, comprising eight males and ten females, with an average age of $24.1 \pm 1.6$ years and they did not have any experience walking uphill or downhill on the treadmill. Their average height and mass were $1.72 \pm 0.09$ m and $67.83 \pm 12.45$ kilograms, respectively. None of the participants had musculoskeletal disorders or a history of joint injuries or replacements that might affect their gait. Individuals self-reporting any other neurological impairments were excluded from participation. Before data collection, participants' dominant leg was identified by asking which leg they preferred to kick a ball; all participants identified their right leg as dominant in this study. This study was carried out with the approval of the University of Nebraska Medical Center Institutional Review Board (IRB# 338-17-FB). Data collection commenced only after participants voluntarily signed the informed consent, and they retained the right to withdraw from the study at any time without providing a reason.

This study adopted a sample size determined by a previously published study examining the impact of plantar vibration on gait characteristics across various inclines (*Xie, Liang & Chien, 2023*). By enrolling 18 healthy young adults, significant effect sizes were observed, with Partial Eta Squared values of 0.4 for step time and 0.127 for step width, signifying substantial effects.

### Experimental protocol

Before the commencement of experimental trials, the preferred walking speed (PWS) of each participant needed to be established. Participants were instructed to stand on the treadmill while holding onto the handrails with both hands. An experimenter then increased the speed of the treadmill belt to 0.8m/s. Participants were asked to walk on the belt for 30 s. Subsequently, the experimenter inquired whether the walking speed felt comfortable, akin to walking on a typical street. Based on participants' feedback, the treadmill speed was either adjusted by increments of 0.1m/s (up or down) or maintained for another 30 s until participants confirmed their PWS. Given the necessity for consistent PWS across the study's various inclines, participants were instructed to experience walking on five different scenarios and select the most comfortable PWS as their final choice. Once the PWS was determined, participants underwent a five-minute familiarization period of walking on a level treadmill at the established PWS. Following the familiarization period,
participants were given a mandatory two-minute rest. Subsequently, participants were randomly assigned to complete five walking trials, each on a different incline (inclines of ±15%, ±8%, and 0% inclination), as outlined in our previous study (*Xie, Liang & Chien, 2023*). The Government of Canada had defined the intensity of inclines (slope gradients) for human walking as follows: 0–3% as little or none, 4–9% as gentle, 10–15% as moderate, 16–30% as steep (https://sis.agr.gc.ca/cansis/nsdb/slc/v3.2/cmp/slope.html#). In the present study, we attempted to investigate the effect of different inclines on the symmetry of walking; thus, we selected gentle (8%) and moderate (15%) walking inclines as the easy and challenging locomotor tasks. Each walking trial lasted two minutes, with a mandatory 2-minute rest period between each trial, as recommended by *Hu & Chien (2021)*. This 2-minute rest period was set to minimize the carryout effect from previous trial (*Hu & Chien, 2021*). Participants were allowed additional rest if necessary to catch their breath before the subsequent walking trial. It's important to note that participants maintained the same PWS regardless of the incline they were walking on.

## Experimental setup

A motion capture system consisting of eight cameras, operating at a sampling rate of 100 Hz (Qualisys AB, Gothenburg, Sweden), was employed to record three-dimensional motion data. Fourteen spherical retro-reflective markers were bilaterally affixed to specific anatomical landmarks, including the anterior superior iliac spine, posterior superior iliac crest, greater trochanter of the femur, lateral epicondyle of the femur, lateral malleoli, toe (second metatarsophalangeal ray), and calcaneal tuberosity. The coordinates data from these marks were extracted using Qualisys Tracker Manager software (Qualisys AB). Subsequently, a customized MATLAB code was used to: (1) filter the raw coordinate data through a zero-lag low pass Butterworth 4th order filter with 10 Hz cut-off frequencies (*Sinclair, Taylor & Hobbs, 2013*); (2) compute the desired outcomes based on a prior study (*Chien, Post & Siu, 2018*). Human walking was conducted on a treadmill (Biodex RTM 600, Shirley, NY, USA) equipped with a safety lanyard. The treadmill's incline was adjustable from 0% to 15%, and it featured an inverse running motor, facilitating downhill walking by altering the walking direction. Participants were provided the same brand (Champion Cross Trainer) running shoes ranging from sizes 6 to 12 and were asked to wear their own socks.

A gait cycle was defined as the duration between two consecutive heel strikes in the ipsilateral leg, with heel strike identified as the point where horizontal heel displacement reached its maximum (*Rose & Gamble, 2006*; *Xie, Liang & Chien, 2023*). Step time was the period between a heel strike on one leg and the next heel strike on contralateral leg, while step length represented the straight-line distance in the sagittal plane between two consecutive heel strikes (Fig. 1A). Step width was the lateral distance between heel markers from one ipsilateral to the subsequent contralateral heel strike (Fig. 1A). Maximum heel clearance referred to the peak height of the heel marker during a gait cycle (Fig. 1B). Time to peak denoted the percentage of a gait cycle at which maximum heel clearance occurred (Fig. 1B). As the customized MATLAB code only measured the horizontal distance or height between heel markers, the calculation of step length and maximum heel clearance

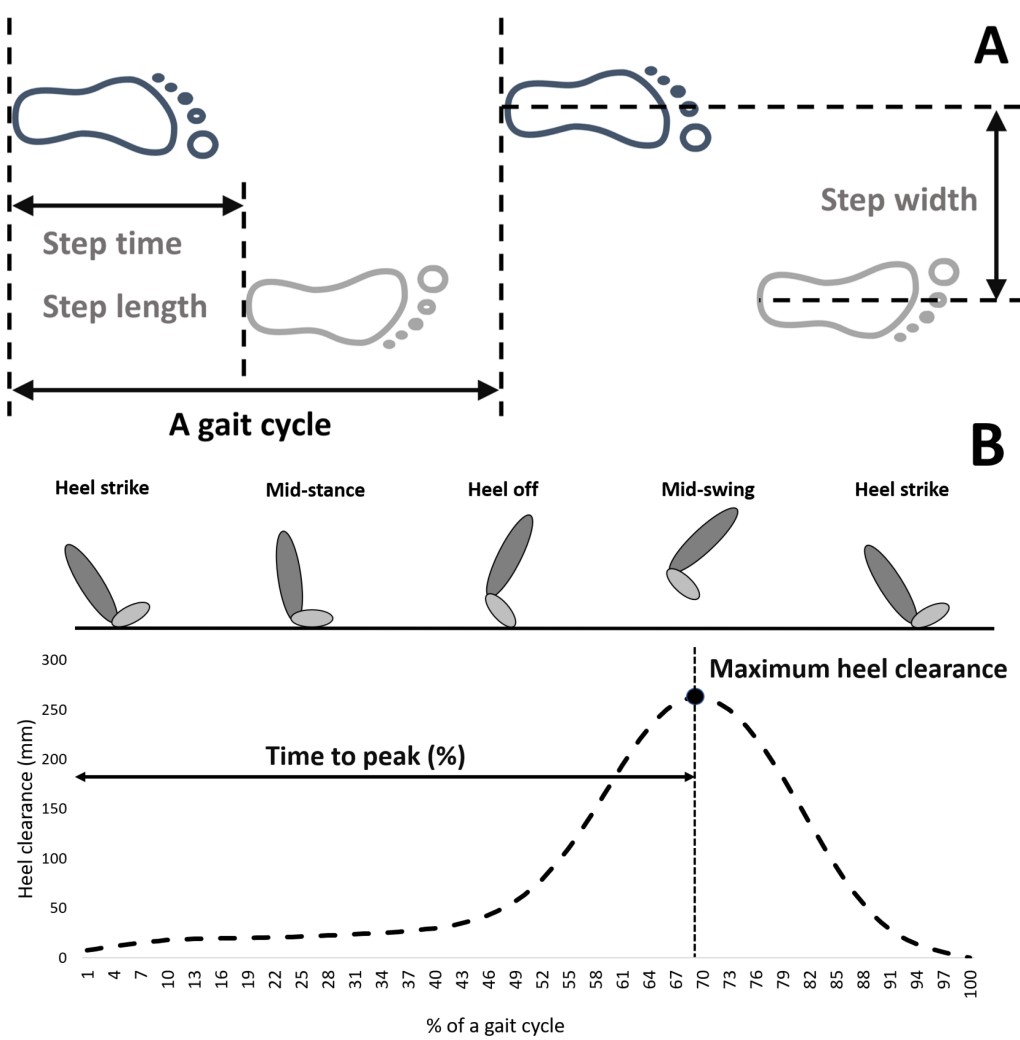

**Figure 1** **Temporal and spatial gait characteristics in this study.** (A) In the anterior-posterior and medial-lateral directions; (B) in the vertical direction. The definition of spatial–temporal gait characteristics in (A) referred to *Rose & Gamble (2006)*. Graph from (B) showing maximum heel clearance was from a gait cycle of a participant in this study while walking on the level treadmill (Data S1).

during uphill and downhill walking needed to take into account the effect of incline, which was presented in Fig. 2. In this study, spatial-temporal parameters were assessed over 100 consecutive gait cycles to mitigate the variability stemming from initial and terminal walking phases (*Xie, Liang & Chien, 2023*). Variability was quantified using the coefficient of variation (CV).

## Statistical analysis

A one-way repeated measures analysis of variance (ANOVA) was used to investigate the effect of five different inclines on each dependent variable. Post-hoc comparisons were performed *via* the Bonferroni correction if a significant difference existed. The significance level was set at .05. Statistical analysis was completed in SPSS 20.0 (IBM Corporation,

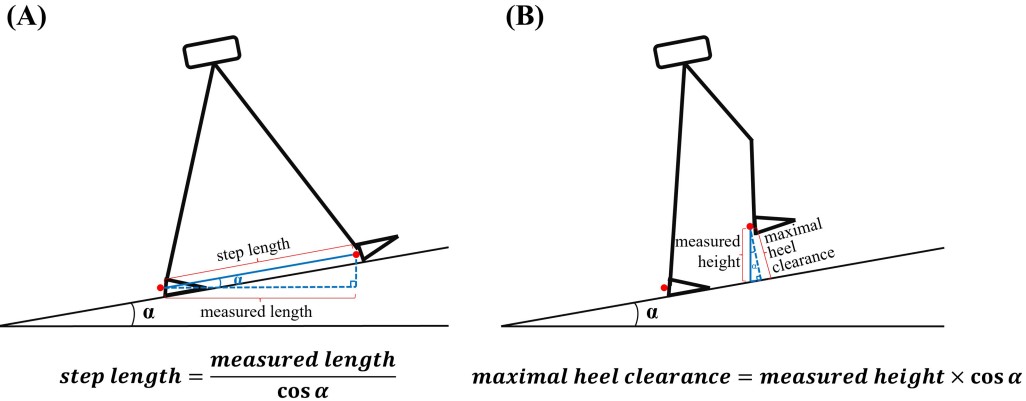

**(A)** **(B)**

$$step\ length = \frac{measured\ length}{cos\ \alpha}$$

$$maximal\ heel\ clearance = measured\ height \times cos\ \alpha$$

**Figure 2 Calculation of step length and maximum heel clearance during uphill and downhill walking in this study.** (A) Step length: Since the customized MATLAB code only measured the horizontal distance between heel markers, step length on uphill and downhill inclines was computed using the formula: measured length/cosine (5.0°) for an 8% incline or measured length/cosine (8.5°) for a 15% incline; (B): maximum heel clearance: Given that the measured height by the MATLAB code represented the vertical distance between heel markers and the treadmill belt, it was adjusted to calculate maximum heel clearance on uphill and downhill inclines using the formula: measured height*cosine (5.0°) for an 8% incline or measured height*cosine (8.5°) for a 15% incline.

Armond, NY). The dependent variables included step time, step length, step width, maximum heel clearance, their variabilities, and respective SIs. The calculation of SI was based on the formula from *Karamanidis, Arampatzis & Brüggemann (2003)*,

$$Symmetry\ index = 2 * \left| \frac{X_D - X_{ND}}{X_D + X_{ND}} \right| * 100 \qquad (1)$$

where $X_D$ and $X_{ND}$ were the values of the dependent variable from the dominant and non-dominant leg, respectively. A greater asymmetry between two legs (reliance on one leg more than another) was indicated by a higher SI value, while a value of 0 suggested perfect symmetry. Applying the absolute function for calculating SI was to prevent that positive and negative deviations cancelled out each other and further hindered the "true" symmetry between two legs (*Karamanidis, Arampatzis & Brüggemann, 2003*). Also, previous studies had determined the preferred leg reliance by using the above formula of SI values in healthy adults (*Stacoff et al., 2005; Van Drongelen et al., 2021*), patients with unilateral hip osteoarthritis (*Schmidt et al., 2017*), patients with bilateral hip osteoarthritis (*Van Drongelen et al., 2021*), and patients with stroke (*Wang & Bhatt, 2022*). Also, available evidence supported that the SI of gait variability could help to evaluate the gait symmetry and investigate respective motor control mechanism comprehensively (*Jochymczyk-Woźniak et al., 2020; Queen et al., 2020*).

## RESULTS

In this study, all participants voluntarily selected PWS when walking on the −15% downhill incline as their final PWS. The average final PWS among participants was 1.08 ± 0.17 m/s. Demographic information from all participants were presented in Table 1.

**Table 1  Summary of demographic information from all participants.**

| Participant No. | Gender | Age (years) | Height (m) | Mass (kg) | Preferred walking speed (m/s) |
|---|---|---|---|---|---|
| 1 | Male | 24 | 1.70 | 61.8 | 1.34 |
| 2 | Female | 25 | 1.93 | 80.3 | 1.34 |
| 3 | Female | 25 | 1.57 | 54.0 | 1.21 |
| 4 | Male | 24 | 1.72 | 65.8 | 1.25 |
| 5 | Female | 23 | 1.70 | 59.4 | 1.25 |
| 6 | Female | 23 | 1.78 | 72.6 | 1.12 |
| 7 | Female | 23 | 1.60 | 45.0 | 1.03 |
| 8 | Male | 23 | 1.82 | 86.2 | 1.12 |
| 9 | Female | 23 | 1.73 | 74.8 | 1.12 |
| 10 | Female | 24 | 1.63 | 52.0 | 0.67 |
| 11 | Female | 24 | 1.83 | 77.1 | 1.03 |
| 12 | Male | 28 | 1.70 | 70.0 | 1.03 |
| 13 | Male | 23 | 1.80 | 74.8 | 1.03 |
| 14 | Female | 23 | 1.68 | 67.1 | 1.25 |
| 15 | Male | 27 | 1.72 | 80.1 | 0.89 |
| 16 | Male | 22 | 1.81 | 87.0 | 1.03 |
| 17 | Female | 26 | 1.73 | 63.5 | 0.89 |
| 18 | Male | 23 | 1.62 | 50.1 | 0.94 |

## Symmetric index of spatial-temporal gait characteristics

Significant differences were observed in the SI of step length ($F_{4,68} = 3.074$, $p = .022$) and maximum heel clearance ($F_{4,68} = 7.047$, $p < .001$). *Post-hoc* pair comparisons showed that both 8% and 15% downhill walking significantly increased SI of step length compared to level walking (−8% *vs.* 0% $p = .011$; −15% *vs.* 0%: $p = .039$). A U shape was observed in the SI of step length (Quadratic, $F_{1,17} = 4.496$, $p = .049$), indicating that no matter increasing or decreasing the inclines from level walking, the SI of step length increased. Also, post-hoc pair comparisons indicated that 8% and 15% downhill walking (−8% *vs.* 0%: $p = .025$; −15% *vs.* 0%: $p = .019$) and 15% uphill walking (15% *vs.* 0%: $p < .001$) decreased the SI of maximum heel clearance compared to level walking. A reverse U shape was observed in the SI of step length (Quadratic, $F_{1,17} = 15.528$, $p < .001$), indicating regardless of an increase or a decrease of the inclines from level walking, the SI of maximum heel clearance decreased. More details were shown in Table 2 and Fig. 3.

## Symmetric index of spatial-temporal gait variabilities

Significant differences were observed in the SI of step length variability ($F_{4,68} = 13.821$, $p < .001$), step width variability ($F_{4,68} = 5.701$, $p = .001$), and maximum heel clearance variability ($F_{4,68} = 2.521$, $p = .049$). As a result of post-hoc pair comparisons, walking on 15% incline resulted in a greater SI of step length variability compared to level walking, and walking on 15% incline resulted in a smaller SI of step width variability compared to level walking. More details are shown in Table 2 and Fig. 4.
**Table 2** Summary of symmetry indexes of dependent variables on different inclines for all participants.

| Inclines | Symmetry index (%) | | | | | | | | | |
|---|---|---|---|---|---|---|---|---|---|---|
| | Step time | Step time variability (CV) | Step length | Step length variability (CV) | Step width | Step width variability (CV) | Maximum heel clearance | Maximum heel clearance variability (CV) | Time to peaks | Time to peaks variability (CV) |
| Declination 15% | 1.9 (1.5) | 9.7 (10.6) | 2.7 (0.7) | 8.8 (6.9) | 0.2 (0.2) | 9.7 (5.3) | 1.6 (1.2) | 6.7 (5.6) | 0.7 (0.5) | 10.9 (7.9) |
| Declination 8% | 1.6 (0.9) | 9.9 (8.0) | 2.7 (0.7) | 12.6 (9.7) | 0.4 (0.7) | 11.5 (6.9) | 1.9 (1.7) | 9.6 (10.0) | 0.7 (0.6) | 13.9 (9.1) |
| Level | 2.2 (1.4) | 18.1 (14.3) | 2.0 (0.5) | 12.1 (7.8) | 0.1 (0.1) | 18.5 (11.5) | 3.3 (1.6) | 9.2 (6.8) | 0.4 (0.4) | 10.1 (5.8) |
| Inclination 8% | 2.6 (1.8) | 14.8 (15.4) | 2.4 (0.9) | 16.6 (10.2) | 0.1 (0.1) | 19.8 (11.0) | 2.6 (2.3) | 14.9 (11.1) | 0.6 (0.5) | 8.7 (8.0) |
| Inclination 15% | 2.6 (1.5) | 11.0 (9.4) | 2.5 (1.1) | 29.5 (13.4) | 0.1 (0.1) | 19.0 (10.9) | 1.3 (1.0) | 14.0 (11.4) | 0.8 (0.7) | 7.7 (5.6) |
| F-value | 1.529 | 1.863 | 3.074 | 13.821 | 2.293 | 5.700 | 7.047 | 2.52 | 1.858 | 2.132 |
| p-value | 0.204 | 0.127 | 0.022 | <0.001 | 0.068 | 0.001 | <0.001 | 0.049 | 0.128 | 0.086 |

**Notes.**
Data are shown as the Mean (Standard deviation). CV, coefficient of variation.

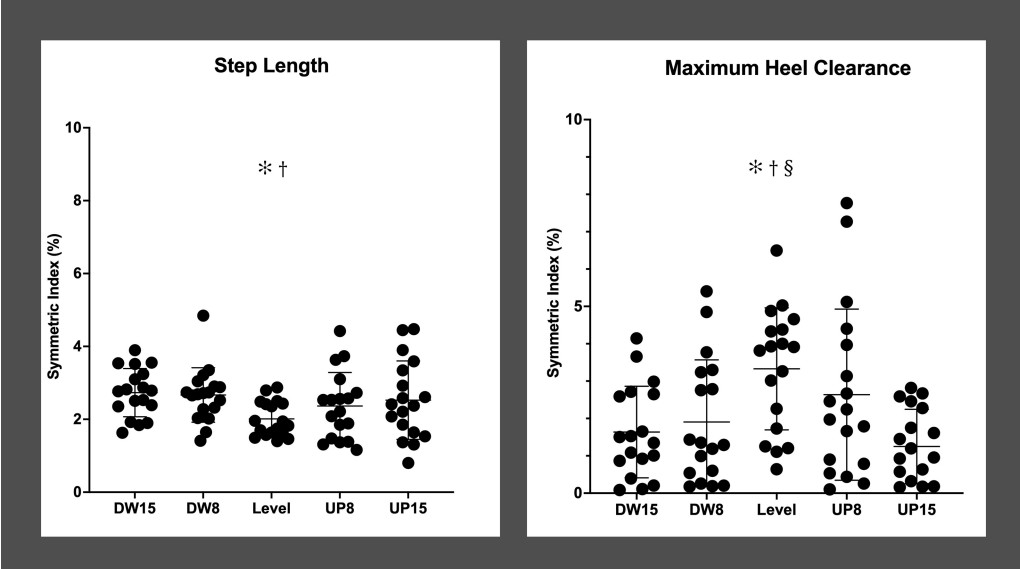

**Figure 3** Symmetry indexes (SI) of step length and maximum heel clearance in 5 different walking inclines. An asterisk (*) indicates the significant difference between −15% downhill incline and other inclines; † indicates the significant difference between −8% downhill incline and other inclines; § indicates the significant difference between 15% uphill incline and other inclines.

## Means of spatial-temporal gait characteristics and its variability

Significant differences among different levels of incline were observed in step length ($F_{4,68} = 92.834$, $p < .001$), step length variability ($F_{4,68} = 15.660$, $p < .001$), step time ($F_{4,68} = 38.379$, $p < .001$), step time variability ($F_{4,68} = 15.665$, $p < .001$), step width ($F_{4,68} = 30.604$, $p < .001$), step width variability ($F_{4,68} = 65.627$, $p < .001$), maximum heel clearance ($F_{4,68} = 190.225$, $p < .001$), maximum heel clearance variability ($F_{4,68} = 24.773$,

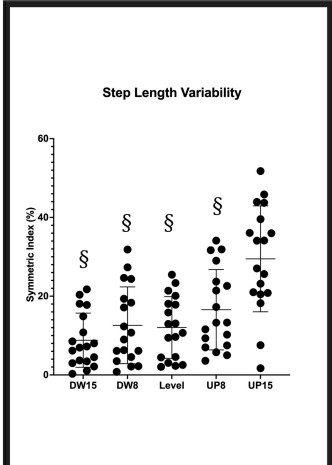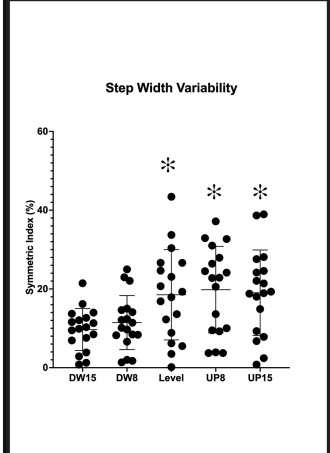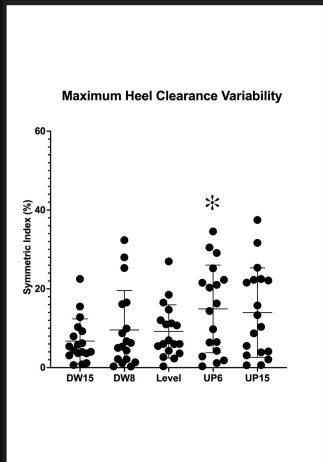

**Figure 4** **Symmetry indexes (SI) of step length variability, step width variability, and maximum heel clearance variability in 5 different walking inclines.** An asterisk (*) indicates the significant difference between −15% downhill incline and other inclines; § indicates the significant difference between 15% uphill incline and other inclines.

**Table 3** **Summary of all dependent variables and respective variabilities on different inclines for all participants.**

| | Declination 15% | | Declination 8% | | Level | | Inclination 8% | | Inclination 15% | |
|---|---|---|---|---|---|---|---|---|---|---|
| | Dominant leg | Non-dominant leg | Dominant leg | Non-dominant leg | Dominant leg | Non-dominant leg | Dominant leg | Non-dominant leg | Dominant leg | Non-dominant leg |
| Step time (ms) | 538.5 (38.9) | 528.1 (35.2) | 559.9 (37.4) | 551.1 (38.2) | 584.2 (52.1) | 571.2 (50.3) | 604.7 (51.1) | 589.3 (55.8) | 609.2 (65.2) | 593.7 (64.5) |
| Step time variability (CV, %) | 3.3 (0.7) | 3.0 (0.7) | 3.2 (1.1) | 2.9 (0.8) | 2.5 (0.8) | 2.1 (0.8) | 2.9 (0.8) | 2.5 (0.9) | 3.3 (1.3) | 3.0 (1.0) |
| Step length (mm) | 667.7 (83.7) | 649.8 (82.4) | 709.6 (92.4) | 691.2 (92.2) | 751.8 (91.7) | 736.7 (89.1) | 783.9 (101.2) | 765.7 (99.7) | 789.1 (107.7) | 769.5 (105.4) |
| Step length variability (CV, %) | 4.6 (1.3) | 4.2 (1.2) | 4.4 (1.5) | 3.8 (1.2) | 3.2 (0.8) | 2.9 (0.7) | 3.8 (1.0) | 3.2 (0.9) | 4.2 (1.0) | 3.2 (0.9) |
| Step width (mm) | 130.5 (29.9) | 130.3 (29.9) | 125.7 (30.6) | 125.4 (30.7) | 201.5 (46.8) | 201.3 (46.8) | 185.9 (35.5) | 185.6 (35.4) | 176.9 (31.0) | 176.8 (31.0) |
| Step width variability (CV, %) | 20.8 (5.2) | 18.9 (4.8) | 20.4 (4.6) | 18.1 (4.1) | 10.4 (5.0) | 8.6 (4.1) | 11.6 (4.9) | 9.4 (3.5) | 11.5 (4.1) | 9.5 (3.2) |
| Maximum heel clearance (mm) | 292.5 (29.6) | 287.8 (29.2) | 276.7 (29.7) | 271.5 (28.7) | 240.5 (24.5) | 232.7 (24.5) | 232.9 (21.2) | 226.9 (20.8) | 234.8 (21.6) | 231.8 (21.2) |
| Maximum heel clearance variability (CV, %) | 3.4 (1.0) | 3.2 (0.9) | 2.5 (0.8) | 2.3 (0.8) | 1.6 (0.6) | 1.5 (0.6) | 2.4 (0.6) | 2.1 (0.5) | 3.0 (0.7) | 2.7 (0.8) |
| Time to peaks (%) | 70.1 (1.7) | 69.6 (1.6) | 71.2 (1.5) | 70.7 (1.3) | 73.5 (1.2) | 73.1 (1.2) | 74.3 (1.2) | 73.9 (1.2) | 75.2 (1.3) | 74.6 (1.3) |
| Time to peaks variability (CV, %) | 1.5 (0.4) | 1.3 (0.3) | 1.3 (0.4) | 1.1 (0.3) | 0.9 (0.2) | 0.8 (0.2) | 1.0 (0.3) | 0.9 (0.2) | 1.2 (0.3) | 1.1 (0.3) |

**Notes.**
Data are shown as the Mean (Standard deviation). CV, coefficient of variation.

$p < .001$), and time to peak of maximum heel clearance ($F_{4,68} = 216.179$, $p < .001$). More details are shown in Table 3.

## The effect size
The effect size was large because the Partial Eta Squared values for the interaction were .484 for SI of step length, .173 for step length variability, and .292 for maximum heel clearance, respectively.

## DISCUSSION

This study investigated the effect of different inclines on symmetry of gait characteristics and respective variabilities among healthy young adults. Also, this attempted to explain these differences in SI of gait characteristics and respective variabilities using internal model hypothesis. The results partially agreed with our hypotheses that (1) walking on inclines affected the SI of step length and maximum heel clearance but not the SI of step time, step width, and time to peak of maximum heel clearance. Specifically, downhill walking significantly increased the symmetry of the maximum heel clearance and decreased the gait symmetry of step length, indicating that downhill walking leaded to a significantly greater preferred leg reliance in comparison to level walking; and (2) for gait variabilities, a significant increase in the symmetry of gait variabilities was observed in step length variability, step width variability, and maximum heel clearance variability.

### Downhill walking significantly decreased the gait symmetry of step length but increased the gait symmetry of maximum heel clearance in comparison to level walking

Our study revealed a U-shaped pattern in the gait symmetry index during both uphill and downhill walking, with downhill walking notably decreasing step length symmetry. This decrease in symmetry might be due to the unfamiliarity of walking on a downhill treadmill (*Darici & Kuo, 2023*). Previous research on split-belt walking aligns with our findings (*Dzewaltowski et al., 2021*; *Mawase et al., 2013*; *Reisman et al., 2007*), suggesting that the forward internal model initiates motor commands based on past experiences with split-belt treadmill. While encountering with an unfamiliar locomotor condition (*e.g.*, the downhill walking), the forward internal model would show a tendency toward the preferred leg to guide step adjustments and maintain balance, which increased the leg reliance as the trade-off. Under the circumstances, the preferred leg may play a leading role in stabilizing the body and determining optimal stepping trajectories. However, it should be noted that the preferred legs might not always be the dominant legs, which can be explained by patients with stroke as an example. Stroke survivors with motor paralysis on their dominant sides commonly rely on intact legs (non-dominant sides) as the preferred ones to compensate for weakness, leading to uneven weight distribution (*Mahon et al., 2015*). Given the gravity-induced effects of a downhill incline, the preferred leg played a critical role in providing sufficient braking power and controlling gait velocity to maintain dynamic balance on the downhill treadmill (*Sample et al., 2020*). *Yang et al. (2019)* further suggested that downhill walking significantly increased negative work (braking power) in the sagittal plane at the hip, knee, and ankle joints to control body descent. Additionally, *Sarvestan et al. (2021)* proposed that the internal model for downhill walking may activate the braking effect to avoid excessive forward propulsion, evidenced by lower knee flexion angles and higher levels of submaximal torque outputs of the quadriceps femoris (*Jeon et al., 2020*). An EMG study conducted by *Alexander & Schwameder (2016)* assessed muscle activity in the lower extremities while walking on inclines of varying degrees. Their findings revealed a significant increase in activation of the rectus femoris during downhill walking compared to walking on level ground, underscoring the crucial role of the knee extensor

in braking and regulating gait velocity when descending slopes (*Alexander & Schwameder, 2016*).

Surprisingly, an inverse U-shaped pattern was observed in the symmetry index of maximum heel clearance, suggesting that walking downhill on the treadmill improved the symmetry of maximum heel clearance. The increase in maximum heel elevation symmetry warranted attention as a trade-off linked to the decrease in step length symmetry. It had been proposed that gait represents an optimization problem, with the overarching objective of reaching a destination with minimal energy expenditure (*Anderson & Pandy, 2001*). This hypothesis was supported by *Nagano et al. (2021)*, which investigated the symmetry of foot-ground clearance in 123 healthy Japanese elders. Their findings suggested that symmetry increased with age, indicating that the aging population enhanced clearance symmetry as a trade-off to manage increased variability (step-to-step adjustment). Although only healthy young adults were recruited in the current study, the increase in symmetry of maximum heel clearance also could be explained as that the improvement of symmetry in the vertical direction was the strategy to handle the downhill walking. Consequently, a couple of summaries may be made: (1) depending on different locomotor tasks, different internal models might be involved to deal with different locomotor challenges; (2) under a challenging locomotor task (downhill walking), the preferred and non-preferred legs might play different roles and were controlled by different internal models separately, in which the preferred leg may be given higher priority over the non-preferred leg; and (3) downhill walking increased the symmetry of maximum heel elevation but decreased the symmetry of spatial gait parameters in the anterior-posterior direction.

### What explained the absence of an effect on the symmetry index (SI) in step width during downhill walking?

Comparing the symmetry index (SI) between anterior-posterior and medial-lateral directions revealed intriguing insights. Downhill walking induced alterations in SI for step length and maximum heel clearance but not for step width. The unchanged medial-lateral gait symmetry during downhill walking suggested two possible explanations. On the one hand, for young adults, downhill walking may not present a sufficient challenge to provoke active adjustments in medial-lateral gait symmetry (*Xie, Liang & Chien, 2023*). It was plausible that healthy young adults primarily modified gait symmetry in the anterior-posterior and vertical directions to navigate inclines safely (*Vieira et al., 2017*). Additionally, during forward downhill walking, anterior-posterior balance control might take precedence over medial-lateral control. This observation aligned with previous findings showing greater gait variabilities in the anterior-posterior direction during lateral stepping based on the lateral active control hypothesis (*Bauby & Kuo, 2000*; *Wurdeman, Huben & Stergiou, 2012*). Bauby and Kuo's active control hypothesis, akin to the internal model concept, suggested that the central nervous system (CNS) received sensory inputs from visual, somatosensory, and vestibular systems to determine the necessity of active motion control based on locomotor tasks. Then, it determined whether active motion control was necessary based on current locomotor tasks. In our study, the internal model initially applied a familiar downhill walking pattern (forward model) and adjusted it based on sensory feedback

(active control). With task repetition, active control decreased, establishing a stable gait pattern (inverse model). Similarly, stroke patients with unilateral lower extremity paralysis demonstrated greater SI of GRF in the anterior-posterior and vertical directions but not in the medial-lateral direction during level walking (*Kim & Eng, 2003*). However, it remained uncertain whether inclined walking further affected gait symmetry in pathological groups. Future research should incorporate pathological groups to explore how diseases influenced the effects of inclined walking on gait symmetry. Understanding how different populations adapted to inclined walking could elucidate unique gait strategies and control mechanisms in various environments (*Dewolf et al., 2021*; *Herssens et al., 2018*; *Kim & Eng, 2003*).

### The symmetry of temporal gait parameters remained unaffected regardless of inclined or level walking

Our findings showed that inclined walking only significantly affected the SI of step length and maximum heel clearance, with no significant differences on temporal gait characteristics or their SIs, regardless of inclines. It aligned with Kimel-Naor, Gottlieb & Plotnik's study (*2017*), which found no differences in temporal gait parameter symmetry during level walking or inclined walking (downhill and uphill) on a self-paced treadmill. The consistent treadmill speed across trials likely contributed to this lack of temporal gait symmetry variation. We proposed that CNS independently regulated temporal and spatial gait parameters, supported by studies involving split-belt treadmills. When walking on such treadmills, participants adapted step lengths to belt speeds without significant changes in step times, suggesting separate regulation of spatial and temporal parameters. This independence implied that spatial gait symmetry adjusted to environmental changes or tasks while temporal symmetry maintained precise step timing. It suggested that foot placement and step timing control were modulated independently during inclined walking (*Gregory, Sup & Choi, 2021*; *Malone & Bastian, 2014*; *Sato & Choi, 2019*).

### Implications for inverse model: why was the larger SI of gait variability identified only when walking on steeper inclines?

There had been a suggestion that the forward model predicted the outcomes of actions, while the inverse model generated actions aiming to achieve desired outcomes (*Pierella et al., 2019*). Therefore, a motor learning could be defined as a concurrent learning of forward and inverse models of actions (*Pierella et al., 2019*). In conjunction with the forward model that had been partially learned, an inverse model could be developed to predict the actions required to achieve anticipative outcomes (*Pierella et al., 2019*). This motor learning process came with the variability. It was important to note that noise (variability) was not only present, but also necessary to facilitate the convergence of learning by minimizing the difference between actual and predicted outcomes and also required higher level of the active control (*Bauby & Kuo, 2000*; *Hu & Chien, 2021*; *Pierella et al., 2019*; *Ren, Lin & Chien, 2022*). In the current study, greater SIs of step length and width variabilities were observed while walking on 15% downhill and uphill inclines, indicating that walking on steeper inclines might require more efforts to convert the forward model to the inverse model even in healthy young adults. Hence, more challenging locomotor tasks might potentially help participants learn more. It was crucial to clarify

that SI of spatial–temporal gait characteristics and variabilities represented two different domains. For gait characteristics, SIs represented the pattern of alterations while SIs of gait variabilities might infer how relative-easy to be learned. Therefore, walking on different inclines could generate a greater SI of gait characteristics but induced a lower SI of gait variability, and vice versa. In other words, downhill walking might help humans more easily transfer the forward model to the inverse model and consistently apply this learned pattern to upcoming walking.

## Clinical applications

This study suggests potential implications for gait rehabilitation, particularly for patients with neurological disorders. Downhill walking, even at a slight incline ($-8\%$), may enhance active control and motor learning, benefiting individuals with impaired gait. Previous research supports this, showing improved gait performance on inclines (*Khandoker et al., 2010*). This study recommends -8% downhill walking training for the elderly or patients with gait pathology, offering a safer and effective rehabilitation protocol.

## Limitations

The present study had the following several limitations. Firstly, the sample size of this study was relatively small. However, according to the measurement of Partial Eta Squared, the effect size of this study was large. Hence, our results demonstrated practical significance. Secondly, only healthy young participants were recruited in this study, which limited the generalization of the current result to other populations. In future research, older adults or patients with neurological diseases, such as stroke and Parkinson's disease, would be included to investigate how aging and pathologic conditions combined with inclined walking affected gait symmetry. Thirdly, in the current study, all participants were required to walk on various inclines to determine their PWS before data collection commenced. This PWS was consistently applied throughout all trials; consequently, only this PWS was recorded, while walking speeds in other conditions were not documented. It was worth mentioning that from our observations, the difference in walking speed across different inclines were minimal, or even identical, among these young healthy adults. However, this may not be the case for the elderly or pathological populations, suggesting that distinct preferred walking speeds might emerge in these groups when traversing different inclines. Therefore, it was essential to conduct further research to investigate whether applying fixed or varying walking speeds influenced gait symmetry in the elderly or pathological individuals. Fourthly, only kinematic data about gait performance was measured in this study. As altered gait kinetics (*e.g.*, muscle activation) leaded to the change in gait kinematics (*e.g.*, spatial-temporal gait characteristics), the observed alterations of kinematics in the current study might come from different levels of muscle activations during level and inclined walking (*Yang et al., 2019*). Future studies would also include the measurement of EMG to investigate how human bodies responded to different inclined walking conditions, as well as the potential association between alterations in gait performance and muscle activations. Fifthly, *Khandoker et al. (2010)* suggested that only when the value of SI was more than 5% be considered as asymmetry. In this study, partial results with significant

differences did not reach 5% of SI because healthy young adults with intact cognitive and sensorimotor functions were recruited and may be less affected by inclined walking. This phenomenon was also observed in the study by *Kaczmarczyk et al. (2017)* that not all SIs of gait variables in healthy young adults were more than 5%. Furthermore, the calculation of SI utilized the absolute values in this study, which removed the ability to detect the effect of dominance on gait symmetry. Future study would consider using another approach, such as the normalized cross-correlation to investigate the gait symmetry (*Ogihara et al., 2020*). Importantly, this study showed the significant differences in SI of gait characteristics caused by inclined walking, confirming that downhill walking enhanced the gait symmetry and the potential clinical implications of using inclined walking to improve gait performance.

## CONCLUSION

In summary, walking on different inclines affected gait symmetry of healthy young adults differently. Particularly, in comparison to level walking, downhill walking significantly improved the gait symmetry in the vertical direction but reduced the gait symmetry in the anterior-posterior direction. The potential mechanism of alterations of gait symmetry might involve that participants utilized different internal models to copy with different challenging locomotor tasks, which ultimately led to the learning of motor skills or patterns. Therefore, the current findings may develop a foundation for future applications of inclined walking on older adults or patients with gait pathology to improve gait symmetry and decrease the risk of falling.

## ACKNOWLEDGEMENTS

We would like to thank all participants for contributing to the study. All data was collected at the Clinical Movement Analysis Lab, Department of Health & Rehabilitation Science at the University of Nebraska Medical Center. We sincerely thank the generosity of the Department of Health & Rehabilitation Science for using the equipment.

### Funding
The authors received no funding for this work.

### Competing Interests
The authors declare there are no competing interests.

### Author Contributions
- Haoyu Xie conceived and designed the experiments, analyzed the data, prepared figures and/or tables, authored or reviewed drafts of the article, and approved the final draft.
- Jung H. Chien conceived and designed the experiments, performed the experiments, analyzed the data, authored or reviewed drafts of the article, and approved the final draft.

## Human Ethics

The following information was supplied relating to ethical approvals (i.e., approving body and any reference numbers):

This study was carried out with the approval of the University of Nebraska Medical Center Institutional Review Board (IRB# 338-17-FB).

## Data Availability

The data is available in the Supplemental File.

## Supplemental Information

Supplemental information for this article can be found online at http://dx.doi.org/10.7717/peerj.18096#supplemental-information.

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
