# Peer review of "Walking on different inclines affects gait symmetry differently in the anterior-posterior and vertical directions: implication for future sensorimotor training"

_PeerJ, doi:10.7717/peerj.18096_

## Round 0.1 · original submission · Minor Revisions

The manuscript follows the correct structure and includes appropriate tables, figures, and raw data. However, several improvements are needed for clarity and accuracy.

Firstly, reword lines 38-39 to specify challenging terrains and clarify the types of patients affected. Add a reference for the statement on stride time variability in line 91. Correct the references for the symmetry index in lines 92-95, and include Robinson (1987) as the original source.

Ensure consistency in terminology for terms like "grade," "incline," and "decline." Clarify whether the same calculation for incline gait was used for decline, as referenced in Figure 2.

Align the hypotheses with the methods and results. Reword the first hypothesis to reflect the measurement of symmetry variability, clarify the second hypothesis on specific comparisons, and either address the third hypothesis in the methods and discussion or remove it if it wasn't tested.

In the discussion, ensure that the results are connected to the stated hypotheses. Add a discussion on limitations, particularly regarding the sample of young, healthy adults and the lack of PWS recording for level and uphill walking. Suggest future research directions to address these limitations.

These revisions are necessary to ensure clarity, consistency, and robustness in presenting the study's aims and findings.

·

Basic reporting

The article is written in clear and professional English.
The introduction presents the context and motivation of the study, highlighting the gaps in the literature on gait symmetry at different inclinations and how this may influence future training.
The article makes appropriate and relevant use of references to substantiate its hypotheses and discuss the results, including classic and recent studies on CPG (Central Pattern Generator) and internal models.
The article structure complies with PeerJ standards and disciplinary norms. The organization facilitates reading and understanding the objectives, methods, results, and discussions.
Figures are relevant, high quality, well labeled, and adequately described.
The raw data was provided following PeerJ policy.

Experimental design

The study is original research and is within the scope of the journal.
The research question is well-defined, relevant, and meaningful. The study addresses a knowledge gap in the existing literature on gait symmetry at different inclinations.
The research was carried out to a high technical and ethical standard, with methods described in sufficient detail for replication.
The study was approved by the Institutional Review Board of the University of Nebraska Medical Center, and all participants signed an informed consent.

Validity of the findings

All underlying data were provided and are robust, statistically sound, and controlled. The use of ANOVA and Bonferroni correction in post-hoc testing is appropriate.
The conclusions are well formulated, linked to the original research question, and limited to the results obtained.
The study concludes that downhill walking increased vertical gait symmetry but reduced anterior-posterior symmetry, suggesting implications for sensorimotor training.

Additional comments

The sample of participants is well-described and adequate for the study.
The methodology is rigorous and detailed, allowing the replication of the study.
The discussion is robust, connecting the findings to CPG concepts and internal models, with clear implications for clinical practice.
The sample includes only young, healthy adults, limiting the generalization of the results to other populations, such as the elderly or individuals with pathologies.
The PWS (Preferred Walking Speed) for level and uphill walking was not recorded, which could offer a more complete comparison.
Consider including a discussion of the limitations mentioned and how future research could address these gaps.

·

Basic reporting

This manuscript follows the correct structure and contains the appropriate tables, figures, and the raw data was shared.

Here are a few recommendations.
Line 38-39: “While healthy adults navigate terrains effortlessly, it’s challenging for toddlers, the elderly, and patients." This should be reworded. Terrains could mean anything, there are plenty of very easy terrains that toddlers and elderly can navigate fine, specify what type of terrains (i.e. challenging, difficult, etc.). Then the word patients is also too broad, there are plenty of patients that do not have their motor control affected and can navigate challenging terrains just fine. Finally, I would qualify that it can be challenging instead of stating that it is challenging.

There are a couple of issues with references in the introduction that need to be corrected.

Line 91: Also, stride time variability … This statement needs a reference.

line 92-95: The references for symmetry index are not correct, additionally the one reference in the methods is not referenced in this section so I would add that one here (Kaczmarczyk, 2017)
• Anders, Balasubramanian, Laroche do not use symmetry index
• Kim & Eng 2003 uses symmetry index proposed by Robinson 1987
• Patterson used SI same as Kim and Eng but also used three other measures to compare.
• Queen tested 4 different ones and proposed a new one, they included SI as one of the 4 but then went on to recommend a new one.
• Wu and Wu mention the symmetry index but then go on to point out it’s limitations and recommend a new formula, the SVM for gait symmetry.

I think SI is fine to use but I would remove the references that don’t use it from this section as well as the 2 references that go on to recommend something else. You might as well reference Robinson 1987 as it appears he was the first to define this metric. [Robinson, et al 1987 Use of force platform variables to quantify the effects of chiropractic manipulation on gait symmetry]

Another area to watch in order to be more clear is the terminology used for grade/incline/decline. In some areas you use the term grade, but then in some you use incline to refer to both uphill and downhill. Make sure you are consistent with terminology.

Experimental design

The experimental design is sound. The data was collected in an appropriate manner and the data was analyzed correctly.

One thing that wasn't clear on the experimental design is the use of the calculation for incline gait seen in figure 2. Was the same calculation done when walking at a decline? This also goes back to clarifying how you are using the terms incline/decline/grade, etc.

Validity of the findings

This paper does a good job of indicating the gap in the literature and the need for the research.

The main problem is the link between the hypotheses, the methods, and the results.

The first hypothesis is “while walking on different inclines (uphill or downhill), significantly greater variability in spatial-temporal gait symmetry will be observed compared to level walking, as even a slight change in incline during treadmill walking may prompt step-by-step adjustments in the internal model.”

Variability of symmetry was never measured in this study, so I think you may mean differences in symmetry of spatiotemporal variability (i.e. the step length variability). If this is the case, then the hypothesis in the introduction just needs to be reworded.

The second hypothesis is “a significant decrease in gait symmetry is anticipated due to alterations in the internal model”

I think you mean a decrease in symmetry when walking at incline and decline compared to walking level following the previous sentences, but I would clarify in this hypothesis statement also.

The third hypothesis is “gait asymmetry predominantly favors the dominant leg”

How was this tested, I don’t see anything in the methods that is testing this hypothesis, and it wasn’t discussed fully in the discussion. Additionally, since your SI calculation takes the absolute value of the dominant-nondominant in the numerator wouldn’t that remove the ability to detect the effect of dominance.

At the beginning of the discussion it is stated that “The results agreed with our hypothesis that walking on different inclines affected the SI of spatial gait characteristics and its variabilities differently”

This hypothesis was not previously stated, and the previously stated hypotheses are not discussed as they are stated.

Overall, this is the biggest concern I have with the paper. There needs to be clearly stated hypotheses, then the methods need to state what measures are being used to test these hypotheses, and then the discussion should discuss how the results support or refute these hypotheses.

Additional comments

Figure 1 depicts the gait cycle including three steps instead of two.

---

## Round 0.2 · accepted · Accept

The authors have addressed all of the reviewers' comments.

·

Basic reporting

The manuscript is well-written, with clear and professional language throughout. The author has successfully addressed the previous feedback by expanding the discussion on the study's limitations, such as the restricted sample size and the lack of EMG data. The references used are current and relevant, providing a strong foundation for the study's hypotheses and conclusions. The inclusion of additional details regarding the internal model and preferred leg reliance has significantly enhanced the manuscript's clarity and depth. Figures and tables are presented in a high-quality format and are well-integrated into the text, further aiding the reader's understanding of the study's results.

Experimental design

The experimental design is robust and well-executed. The author clearly describes the methods used, including the identification of participants' preferred walking speed (PWS) and the experimental protocol for walking on various inclines. The study design is appropriate for addressing the research questions posed, and the inclusion of a detailed statistical analysis plan (ANOVA with Bonferroni correction) strengthens the reliability of the findings. While the absence of EMG data was noted as a limitation, the author has acknowledged this and suggested it as a direction for future research, which is commendable.

Validity of the findings

The findings presented in the manuscript are valid and well-supported by the data. The author has effectively addressed previous concerns regarding the interpretation of the symmetry index (SI) and has provided a more nuanced discussion of the results, particularly in the context of different inclines affecting gait symmetry in varying directions (anterior-posterior vs. vertical). The manuscript provides a thorough analysis of the impact of incline on gait symmetry and variability, with significant differences clearly reported and interpreted. The discussion has been enhanced with additional references and a more comprehensive exploration of the internal model's role in gait control, lending greater validity to the conclusions drawn.

Additional comments

The revisions made by the author have substantially improved the manuscript. The discussion now includes a thoughtful analysis of the study's limitations and suggests future research directions that could address these gaps. The enhanced discussion on the internal model and its implications for gait symmetry is particularly valuable and adds depth to the study's contributions to the field. The manuscript is now well-positioned for publication, with only minor editorial adjustments needed to ensure clarity and consistency throughout the text.

·

Basic reporting

The article is well written and uses proper English, The structure, figures, and tables are all appropriate. All concerns from the first review were addressed sufficiently.

Experimental design

The clarification of the experimental hypothesis greatly improved the overall article. There are no further concerns about the experimental design.

Validity of the findings

There are no concerns with the validity of the findings. The data was provided in an easily accessible format.